# Effect of Continuous Positive Airway Pressure on Changes of Plasma/Serum Ghrelin and Evaluation of These Changes between Adults with Obstructive Sleep Apnea and Controls: A Meta-Analysis

**DOI:** 10.3390/life13010149

**Published:** 2023-01-04

**Authors:** Amin Golshah, Mohammad Moslem Imani, Masoud Sadeghi, Mozhgan Karami Chalkhooshg, Annette Beatrix Brühl, Laleh Sadeghi Bahmani, Serge Brand

**Affiliations:** 1Department of Orthodontics, Kermanshah University of Medical Sciences, Kermanshah 6715847141, Iran; 2Department of Biology, Science and Research Branch, Islamic Azad University, Tehran 1477893855, Iran; 3Students Research Committee, Kermanshah University of Medical Sciences, Kermanshah 6715847141, Iran; 4Center for Affective, Stress and Sleep Disorders (ZASS), Psychiatric University Hospital Basel, 4002 Basel, Switzerland; 5Department of Education and Psychology, Shahid Ashrafi Esfahani University, Esfahan 1461968151, Iran; 6Sleep Disorders Research Center, Kermanshah University of Medical Sciences, Kermanshah 6715847141, Iran; 7Department of Sport, Exercise and Health, Division of Sport Science and Psychosocial Health, University of Basel, 4052 Basel, Switzerland; 8Substance Abuse Prevention Research Center, Kermanshah University of Medical Sciences, Kermanshah 6715847141, Iran; 9School of Medicine, Tehran University of Medical Sciences, Tehran 1417613151, Iran; 10Center for Disaster Psychiatry and Disaster Psychology, Psychiatric University Hospital Basel, 4002 Basel, Switzerland

**Keywords:** sleep apnea syndromes, ghrelin, serum, plasma, meta-analysis

## Abstract

Background and objective: Obstructive sleep apnea (OSA) can be related to high ghrelin hormone levels that may encourage additional energy intake. Herein, a new systematic review and meta-analysis were performed to check the changes in serum/plasma levels of ghrelin in adults with OSA compared to controls, as well as before compared after continuous positive airway pressure (CPAP) therapy in adults with OSA. Materials and methods: Four main databases were systematically and comprehensively searched until 17 October 2022, without any restrictions. For assessing the quality, we used the Joanna Briggs Institute (JBI) critical appraisal checklist adapted for case–control studies and the National Institutes of Health (NIH) quality assessment tool for before–after studies. The effect sizes were extracted by the Review Manager 5.3 software for the blood of ghrelin in adults with OSA compared with controls, as well as before and after CPAP therapy. Results: Fifteen articles involving thirteen studies for case–control studies and nine articles for before–after studies were included. The pooled standardized mean differences were 0.30 (95% confidence interval (CI): −0.02, 0.61; *p* = 0.07; I^2^ = 80%) and 0.10 (95% CI: −0.08, 0.27; *p* = 0.27; I^2^ = 42%) for case–control and before–after studies, respectively. For thirteen case–control studies, nine had moderate and four high qualities, whereas for nine before–after studies, five had good and four fair qualities. Based on the trial sequential analysis, more studies are needed to confirm the pooled results of the analyses of blood ghrelin levels in case–control and before–after studies. In addition, the radial plot showed outliers for the analysis of case–control studies that they were significant factors for high heterogeneity. Conclusions: The findings of the present meta-analysis recommended that the blood levels of ghrelin had no significant difference in the adults with OSA compared with the controls, nor did they have significant difference in adults with OSA before compared with after CPAP therapy. The present findings need to be confirmed in additional studies with more cases and higher qualities.

## 1. Introduction

Obstructive Sleep Apnea (OSA) is a state determined by repeated episodes of partial or complete airway obstruction over sleep [1,2]. Apnea–Hypopnea Index (AHI) or the number of apneas/hypopneas per hour of sleep is the basic metric for identifying OSA that this is evaluated by Polysomnography (PSG) or other shapes of sleep monitoring [3]. The overall prevalence of OSA in adults (AHI ≥ 5 events/h) ranged from 9–38% in the public adult population and was more in men [4]. In addition, the OSA prevalence was calculated to be 56.0% in patients with type 2 diabetes [5].

Obesity, smoking, alcohol consumption, higher age, and male gender can be the risk factors for OSA [6]. It has been shown the impact of ethnicity on the prevalence and severity of OSA that this impact can be related to ethnic differences in adipose tissue distributions [7]. Apart from the environmental and demographical factors, the studies reported that genetic [8,9,10,11] and blood [12,13,14,15,16,17,18,19] factors could also affect the prevalence or development of OSA.

Ghrelin (a 28 amino acid hormone or orexigenic neuropeptide involving an n-octanoyl group on the serine in position 3) [20] is known as an endocrine pathway in controlling nutrition and energy balance that is secreted by a large number of tissues, but its dominant source is the gastric mucosa [21,22]. Ghrelin is in both acylated and unacylated forms [23]. Acylated ghrelin is the active shape of ghrelin [24] (acylation is essential for the ghrelin binding and function [25]) with some metabolic functions such as appetite stimulation, reduced insulin secretion from pancreatic, elevated growth hormone secretion, reduced body energy consumption, and environmental growth and metabolism, especially carbohydrates and fats [22].

The OSA is related to hormonal features and is illustrated by high levels of ghrelin and leptin hormones that may provoke additional energy intake [26]. In men with OSA, energy expenditure relative to body weight reduces with elevating severity of oxygen desaturation that can contribute to a positive energy balance [27]. The studies [28,29] reported different results for blood levels of ghrelin in adults with OSA in comparison with controls. The relationship between OSA and plasma/serum levels of ghrelin is controversial [30]. Obesity [31], cardiovascular diseases [32], diabetes and metabolic syndrome [33], and hypertention [34] are associated with blood ghrelin levels and on the other hand, OSA is related to these disorders or diseases [35,36,37]. Therefore, finding a link between ghrelin levels with OSA development can be useful for prediction of related diseases with OSA and possible treatments. 

The OSA cases treated by nasal Continuous Positive Airway Pressure (CPAP) require using CPAP therapy to stop the recurrence of symptoms [38]. The changes in energy metabolism accrue after CPAP therapy for OSA [39]. The studies [40,41,42] reported the impact of CPAP therapy on the blood levels of ghrelin in adults with OSA with different results. 

Based on our knowledge of English literature, there was a meta-analysis [30] related to this subject—searching three databases until 2018—with eight case–control and six before–after studies. Therefore, a new systematic review and meta-analysis in four main databases were conducted with more studies (thirteen case–control and nine before–after studies) and additional analyses than the previous meta-analysis for findings potentially effective factors on heterogeneity and bias (radial plot analysis, meta-regression, and trial sequential analysis (TSA)) to check the changes of serum/plasma levels of ghrelin in adults with OSA compared to controls, as well as before compared after CPAP therapy in adults with OSA with more details. In addition to a few new studies, the previous meta-analysis missed several articles before 2018 that could be due of the choices of database or the searching criteria.

## 2. Materials and Methods

To design of the present meta-analysis, it was followed the PRISMA-P items [43]. The PECO question [44,45] was: Are blood ghrelin levels different in adults with OSA in comparison to controls? (P: human adults with and without OSA, E: OSA disorder, C: adults with OSA compared to controls; O: and the plasma/serum ghrelin level). The clinical PICO (Population, Intervention, Comparator, and Outcome) question was: What is the impact of CPAP therapy on serum/plasma levels of ghrelin in adults with OSA? (P: human adults with OSA, I: CPAP therapy, C: adults with OSA before and after CPAP therapy; O: and the plasma/serum ghrelin level).

### 2.1. Search Strategy

Four databases (PubMed, Web of Science, Scopus, and Cochrane Library) were systematically and comprehensively searched until 17 October 2022, without any restrictions by one reviewer (M.S.). The search terms were as: (“obstructive sleep apnea” or “sleep apnea” or “OSA” or “obstructive sleep apnea syndrome” or “OSAS” or “obstructive sleep apnea-hypopnea syndrome” or “OSAHS”) and (“ghrelin”). The citations of all types of articles linked to the subject and “Google Scholar” were checked to ensure no study was missed. 

### 2.2. Eligibility Criteria

Inclusion criteria: (1) studies including both adults with OSA and controls aged ≥18 years without any treatment or adults with OSA under CPAP therapy, (2) studies reporting plasma/serum ghrelin levels in OSA and controls or adults with OSA before and after CPAP therapy, (3) PSG was applied to diagnose OSA, defined as AHI ≥5 events/h for adult, (4) adults with OSA did not have other systemic diseases (diabetes mellitus, cardiovascular diseases, heart, hepatic, and renal failures, and lung diseases, any malignancy, and infectious diseases, other sleep disorders), (5) controls did not have OSA or systemic disease (see the previous criterion), and (6) venous blood was took in the fasting state on the morning to measure ghrelin. Exclusion criteria: (1) meta-analyses, book chapters, conference papers, the letter to the editor, commentary, and reviews, (2) studies without complete data, (3) studies in the absence of a control group or the control group had AHI was more than 5 events/h, (4) studies including participants aged less than 18 years old, and (5) studies including adults with OSA with any another disease.

### 2.3. Data Collection

The data were extracted for any study involved in the meta-analysis by two independent reviewers (A.G. and M.S.). The differences between reviewers were resolved by third reviewer (S.B.). Extracted data were the country and ethnicity of participants, the first author, the publication year, ghrelin sampling, the sample size of adults with OSA and controls, quality or quality score, means BMI, age, and AHI the groups, follow-up duration of CPAP therapy, mean AHI before and after CPAP therapy, and mean of blood levels of ghrelin in all groups. 

### 2.4. Quality Assessment

For assessing the quality, we used the Joanna Briggs Institute (JBI) critical appraisal checklist adapted for case–control studies including ten questions or ten scores as Low: 1–4 scores, Moderate: 5–7 scores, High: 8–10 scores [46] and the National Institutes of Health (NIH) quality assessment tool for before–after studies with twelve question or twelve scores as Good: 9–12 scores, Fair: 5–8 scores, Poor: 1–4 scores [47] (See Appendix A). The quality score were performed by two independent reviewers (M.M.I. and M.S.). The differences between reviewers were resolved by third reviewer (M.K.C.).

### 2.5. Statistical Analyses

The Review Manager 5.3 (RevMan 5.3) software was applied to extract the effect sizes (standardized mean difference (SMD) and 95% confidence interval (CI)) of blood levels of ghrelin amongst adults with OSA and controls, as well as before and after CPAP therapy by one reviewer (M.S.). The *p*-value (2-sided) of less than 0.05 was considered a significant value. A P_heterogeneity_ < 0.1(I^2^ > 50%) reported a significant heterogeneity that in this state, a random-effects model [48], otherwise, a fixed-effect model [49] was used. 

The subgroup and random-effect meta-regression analyses were done based on several variables and evaluating the stability of initial pooled SMDs, both “one-study-removed” and “cumulative” analyses as sensitivity analyses were utilized. 

The Begg’s funnel plot by Begg’s test was applied to test potential publication bias [50] and the Egger’s test to report degree of asymmetry [51] that the *p*-values of both tests and the data for sensitivity analyses were extracted by the Comprehensive Meta-Analysis version 2.0 (CMA 2.0) software and a *p*-value (2-sided) less than 0.10 recommended the existence of the publication bias. 

To report the potential random error (false-positive and -negative results) in meta-analysis [52], trial sequential analysis (TSA) was accomplished using TSA software (version 0.9.5.10 beta) [53]. The futility threshold can show a no-impact result before attaining the information size. An α-risk of 5%, a β-risk of 20%, and a 2-sided border type reporting the mean difference and variance were based on empirical assumptions created automatically by the software, were used to calculate the required information size (RIS). If the Z-curve reached the RIS line, enough participants were included in the studies and the conclusion was trustworthy or crossed the borderlines the results could be robust. Differently, the volume of information was not large enough and more evidence was needed.

The effect sizes for the studies including the required data just on a graph were extracted from the graph utilizing GetData Graph Digitizer 2.26 software. 

## 3. Results

### 3.1. Search Strategy

To search in the databases, 362 records were identified and after deleting duplicates and irrelevant records, 28 full-text articles were evaluated (Figure 1). Then, 13 articles were excluded with reasons (one was a meta-analysis, three were reviews, two did not report a control group or adults with OSA under CPAP therapy, two had no relevant data, two included a control group with AHI >5 events/h, one reported geometrical data, and two were reported in children). At last, 15 articles involving 13 studies for case–control studies and 9 articles for before–after studies were included. All studies reported total ghrelin levels except one study [42] that reported acylated ghrelin. 

### 3.2. Characteristics of the Studies

Based on fifteen articles [28,29,39,40,41,42,54,55,56,57,58,59,60,61,62], Table 1 and Table 2 display the characteristics of the case–control and before–after studies in the analysis, respectively. With regard to thirteen case–control studies, nine studies were reported in Caucasians, three in Asians, and one in a population with mixed ethnicity, whereas in nine before–after studies, four in Caucasians, four in Asians, and one in a population with mixed ethnicity. In case–control studies, seven studies reported plasma levels of ghrelin and six serum levels, whereas in before–after studies, six plasma levels and three serum levels. Data of other variables such as sample size, mean BMI, mean age, mean AHI, and follow-up duration are reported in Table 1 and Table 2.

### 3.3. Pooled Analyses

Figure 2 and Figure 3 show the forest plot analyses of blood ghrelin levels in adults with OSA in comparison to controls and adults with OSA before and after CPAP therapy, respectively. The pooled SMDs were 0.30 (95% CI: −0.02, 0.61; *p* = 0.07; I^2^ = 80%) and 0.10 (95% CI: −0.08, 0.27; *p* = 0.27; I^2^ = 42%) in case–control and before–after studies, respectively. Therefore, the results recommended that there were no significant differences between adults with OSA and controls, moreover between adults with OSA before and after CPAP therapy (there was no effect of CPAP therapy on OSA).

### 3.4. Quality Scores

Table 3 and Table 4 show JBI critical appraisal checklist for case–control studies and NIH quality assessment tool for before–after studies, respectively. The questions of the JBI critical appraisal checklist and the NIH quality assessment tool have been reported in the Appendix A. Of thirteen case–control studies, nine had moderate and four had high qualities. Of nine before–after studies, five had good and four had fair qualities.

### 3.5. Subgroup Analyses

Table 5 and Table 6 report the subgroup analyses (for finding probably effective factors for heterogeneity) for the case–control and before–after studies, respectively. The subgroup analysis based on ethnicity, blood sample, sample size, mean BMI of adults and mean age of adults with OSA and controls, mean AHI of adults with OSA, and quality as important factors in adults with OSA and also checked factors in most studies were checked for case–control studies. The results showed that blood sample, sample size, mean age of adults with OSA, and quality were effective factors for the pooled analysis of the blood levels of ghrelin in adults with OSA in comparison to controls, as well as heterogeneity across the studies. In addition, the subgroup analyses based on ethnicity, blood sample, sample size, mean BMI, mean age, mean AHI of adults with OSA before treatment, and quality were checked for before–after studies. The results showed that just mean AHI before treatment was an effective factor for the poled analysis of blood levels of ghrelin in adults with OSA before in comparison to after CPAP therapy. 

### 3.6. Meta-Regression Analyses

Table 7 and Table 8 include the data of the meta-regression analyses for the case–control and before–after studies, respectively. The results represented that publication year, sample size, mean AHI of adults with OSA, and quality were confounding factors for the blood levels of ghrelin in adults with OSA versus controls (increasing publication year and sample size, the level of ghrelin significantly increased, but increasing mean AHI of adults with OSA and quality score, the level of ghrelin significantly decreased. Among the factors checked in before–after studies, there was no confounding factor. Therefore, publication year, sample size, mean AHI of adults with OSA, and quality can affect the ghrelin levels and these factors can be probably effective factors for heterogeneity across the studies. 

### 3.7. Radial Plots

Figure 4 identifies the radial plots for the case–control and before–after studies, respectively. The radial plot confirmed the high heterogeneity between the case–control studies. Two studies [28,57] were outliers and removing these studies as a sensitivity analysis, there was a lack of heterogeneity. Therefore, outliers are a significant effective factor for high heterogeneity between the studies. The radial plot confirmed that there was no heterogeneity due to outliers for before–after studies. 

### 3.8. Sensitivity Analyses

The sensitivity analyses reported stability of the pooled results for both case–control and before–after studies. There were two outliers [28,57] for the case–control studies that removed them, pooled SMD became 0.22 (95% CI: 0.05, 0.39; *p* = 0.010; I^2^ = 0%). The result showed a significantly high level of ghrelin in adults with OSA vs. controls.

### 3.9. Trial Sequential Analyses (TSAs)

Figure 5 shows TSAs for the case–control and before–after studies. The result of TSA showed that the cumulative Z-curve crossed both the conventional boundary and the trial sequential monitoring boundary, which recommended that the result of the analysis of blood ghrelin level in adults with OSA vs. controls is robust with 1009 cases. Although the actual sample size did not exceed the RIS of 1280 cases, therefore definite result could not be obtained for this analysis, and more studies with sufficient evidence are needed. The results did not confirm the sufficient cases and evidence for the pooled analyses of the blood levels of ghrelin in before–after studies. Because Z-curve did not cross the RIS line or monitored the boundaries in the analysis of before–after studies. Therefore, more studies with sufficient evidence are needed in the future to confirm this result of the analysis of blood ghrelin levels before compared with after CPAP therapy in adults with OSA.

### 3.10. Publication Bias

Figure 6 represents the funnel plots of serum/plasma ghrelin levels in both case–control and before–after studies. The results display no publication bias among case–control (*p*-values: Egger’s = 0.357 and Begg’s = 0.714) and before–after (*p*-values: Egger’s = 0.891 and Begg’s = 0.834) studies.

## 4. Discussion

The relationship between OSA and plasma/serum ghrelin levels and the effect of CPAP therapy on ghrelin levels have remained controversial [30]. The main results of the present meta-analysis recommended that the serum/plasma levels of ghrelin had no significant difference in the adults with OSA compared to the controls, moreover in adults with OSA before compared to after CPAP therapy. Removing outliers, the serum/plasma levels of ghrelin were significantly higher in the adults with OSA compared to the controls. Two analyses included low sample sizes based on TSA results. Blood sample, sample size, quality scores, means age, and AHI of adults with OSA were effective factors in case–control studies, and the mean AHI of adults with OSA before CPAP therapy in before–after studies. Therefore, the present findings require to be confirmed in additional studies with more cases and higher qualities.

Among thirteen case–control studies, three studies [28,41,61] showed a significantly high level of ghrelin, whereas other studies did not find any significant difference between in adults with OSA versus controls. Among all before–after studies in the present meta-analysis, the CPAP therapy had a significant defect in increasing [40] and decreasing [41] the blood levels of ghrelin, but other studies did not find any effect of CPAP on the levels of ghrelin in adults with OSA.

A systematic review recommended the positive impact of older age, male gender, and higher BMI on OSA prevalence [4]. Research showed that plasma ghrelin decreased in obese people and increased in lean people [63]. Ciftci et al. [55] revealed that serum ghrelin level has a positive correlation with BMI and AHI. Other studies confirmed the positive correlation of serum ghrelin levels with BMI [40] and AHI [60]. However, a number of studies did not confirm the correlation of serum ghrelin level with BMI [56] and AHI [40,56]. Whatever the present meta-analysis showed the correlation of AHI and age with blood ghrelin levels in adults with OSA, but it did not find any significant correlation between blood levels of ghrelin and BMI. 

One study [28] reported a significant association between serum ghrelin levels and the severity of OSA as serum level of ghrelin was significantly higher in adults with severe OSA vs. moderate OSA and moderate OSA compared to mild OSA. Unfortunately, most studies did not report the blood levels of ghrelin based on OSA severity and therefore we could not analyze the association between the blood ghrelin levels and the severity of OSA. The researchers need to perform this analysis among adults with OSA in their original articles in the future. In addition, results of this current meta-analysis were in line with the previously published meta-analysis [30]. 

There were three significant limitations during the meta-analysis design. (1) A low number of participants in the studies and low included studies in each analysis. (2) Less number of studies had high quality. (3) High heterogeneity among case–control studies. In contrast, there were two important strengths. (1) The stability of results. (2) A lack of publication bias across the studies.

## 5. Conclusions

The present meta-analysis recommended that the blood levels of ghrelin had no significant difference in the adults with OSA vs. the controls, moreover in adults with OSA before vs. after CPAP therapy. Notwithstanding the low number of individuals in the analyses, the study reported that blood sample, sample size, quality scores, mean age, and mean AHI of adults with OSA were effective factors in case–control studies, and mean AHI of adults with OSA before CPAP therapy in before–after studies. Therefore, the present findings require to be accepted in additional studies with more cases and higher qualities.

## Figures and Tables

**Figure 1 life-13-00149-f001:**
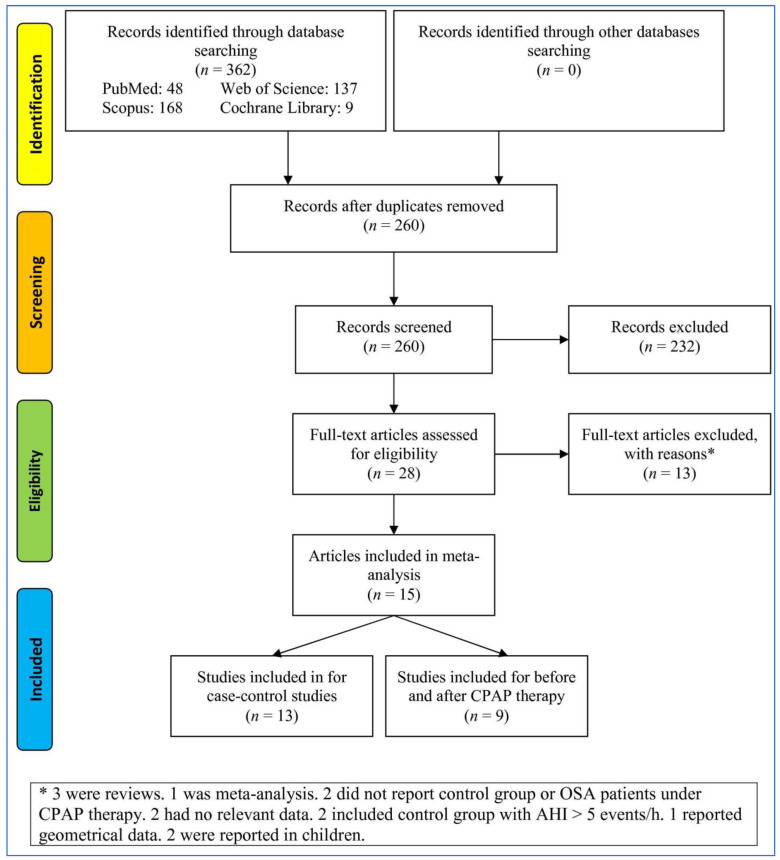
Flowchart of the study selection. CPAP: Continuous positive airways pressure. AHI: Apnea–hypopnea index. OSA: Obstructive sleep apnea.

**Figure 2 life-13-00149-f002:**
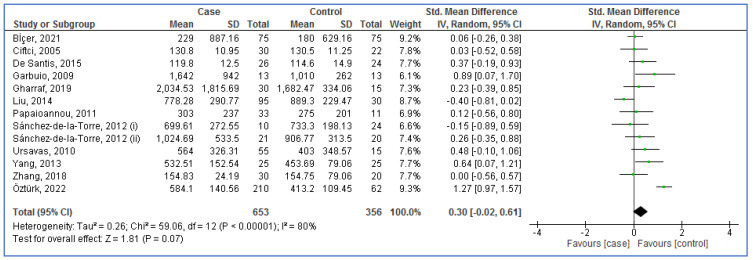
Forest plot analysis of serum/plasma ghrelin levels in adults with obstructive sleep apnea versus controls. The diamond at the bottom of the forest plot represents the result when all the individual studies are combined together and averaged. Names of studies are shown on the left, std. mean differences (green boxes) and confidence intervals (horizontal lines) on the right. The left column shows the first author’s names and publication years of studies for twelve arti-cles [28,29,40,41,54,55,56,57,58,59,60,61] included in the analysis. One article [59] included two independent studies marked with i and ii.

**Figure 3 life-13-00149-f003:**
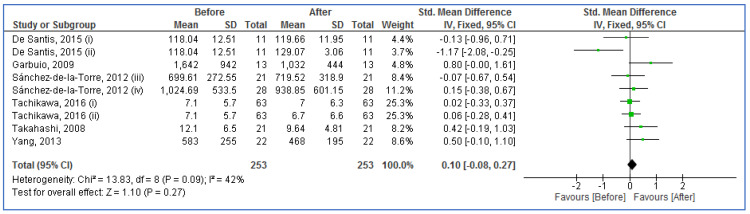
Forest plot analysis of serum/plasma ghrelin levels in adults with obstructive sleep apnea before and after continuous positive airways pressure therapy. The diamond at the bottom of the forest plot represents the result when all the individual studies are combined together and averaged. Names of studies are shown on the left, std. mean differences (green boxes) and confidence intervals (horizontal lines) on the right. The left column shows the first author’s names and publication years of studies for six articles [39,40,41,42,59,62] included in the analysis. three articles [39,40,59] included two independent studies each one marked with i and ii or iii and iv.

**Figure 4 life-13-00149-f004:**
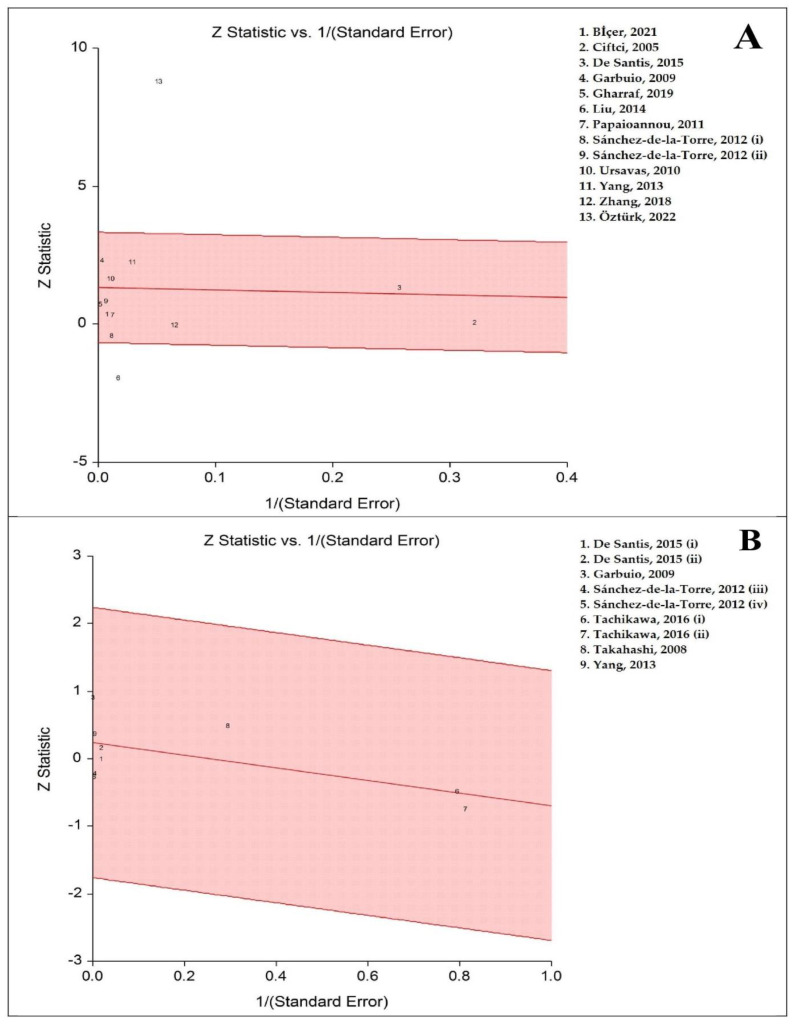
Radial plots of serum/plasma ghrelin levels. (**A**) Adults with obstructive sleep apnea vs. controls. (**B**) Before and after continuous positive airways pressure therapy in adults with OSA. Each number shows one study that the studies with related numbers are represented in right side. Each number in up and right shows one study that (**A**) shows thirteen studies from twelve articles [28,29,40,41,54,55,56,57,58,59,60,61] and (**B**) ahows nine studies from six articles [39,40,41,42,59,62]. The articles with two independent studies [39,40,59] have marked with i and ii or iii and iv.

**Figure 5 life-13-00149-f005:**
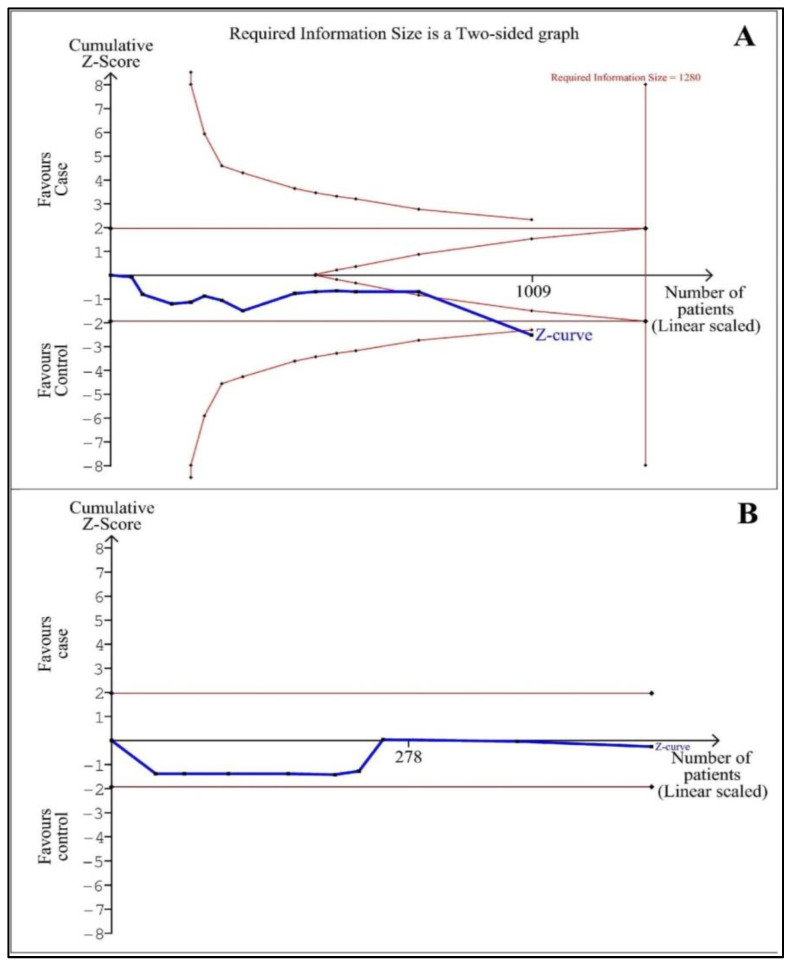
Trial sequential analysis of serum/plasma ghrelin levels. (**A**) Adults with obstructive sleep apnea compared to controls (D^2^ = 98%). (**B**) Before and after continuous positive airways pressure therapy (D^2^ = 81%) in adults with OSA. The red horizontal lines show monitoring boundaries for benefit (upper line), monitoring boundaries for harm (lower line), and futility boundaries (middle lines). The red vertical line is related to the required sample size.

**Figure 6 life-13-00149-f006:**
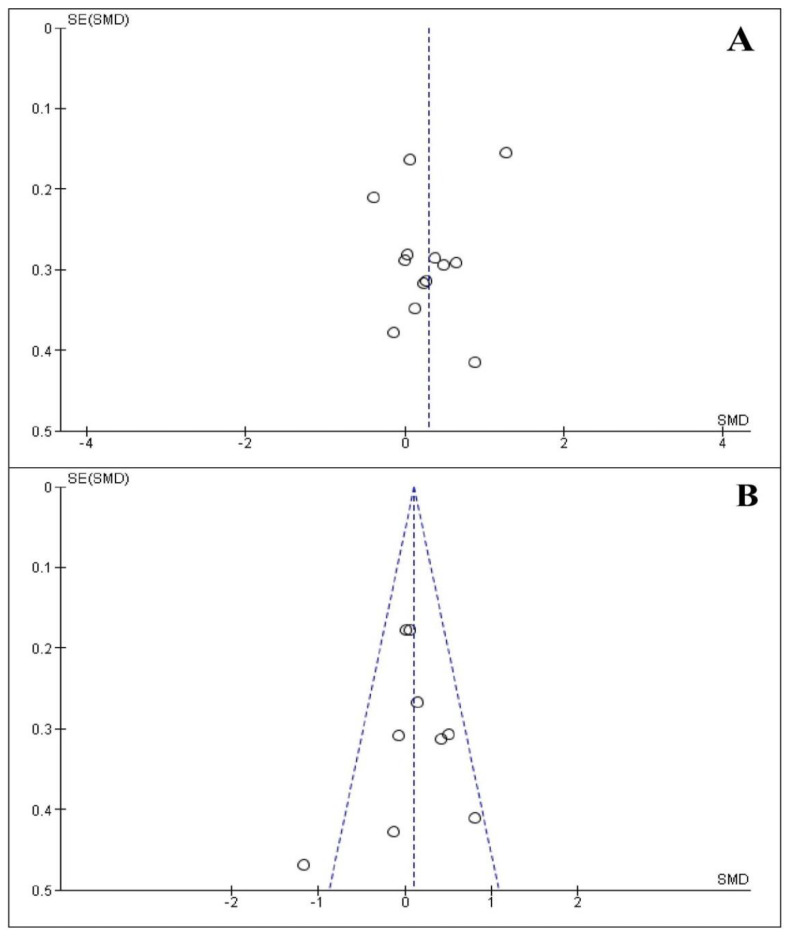
Funnel plots of serum/plasma ghrelin levels. (**A**) Adults with obstructive sleep apnea compared to controls. (**B**) Before and after continuous positive airways pressure therapy in adults with OSA. SMD: Standardized mean difference. SE: Standard error. Circles represent individual studies. The diagonal dashed lines represent the pseudo 95% confidence intervals around the pooled SMD for each standard error of the ordinate vertical axis values. The vertical dashed line represents the pooled SMD.

**Table 1 life-13-00149-t001:** Characteristics of the case–control studies in the meta-analysis.

First Author, Publication Year	Country	Ethnicity	Sample Size (Case/Control)	Mean BMI, kg/m^2^		Mean Age, Year		Mean AHI, Events/h		Sample
				Case	Control	Case	Control	Case	Control	
De Santis, 2015 [40]	Italy	Caucasian	26/24	33.0	38.0	41.8	43.7	26.15	1.65	Serum
Liu, 2014 [57]	China	Asian	95/30	28.45	27.85	47.57	45.35	30.28	3.07	Plasma
Ciftci, 2005 [55]	Turkey	Caucasian	30/22	32.12	31.03	Matched	Matched	44.24	1.55	Serum
Yang, 2013 [61]	China	Asian	25/25	27.5	26.22	53	54	25	3	Plasma
Zhang, 2018 [29]	China	Asian	30/20	28.85	27.55	40.73	36.10	61.48	1.93	Plasma
Sánchez-de-la-Torre, 2012 (i) [59]	Spain	Caucasian	10/24	34.34	32.01	46.61	48.7	48.92	2.87	Plasma
Sánchez-de-la-Torre, 2012 (ii) [59]	Spain	Caucasian	21/20	25.02	24.71	49.33	42.9	41.45	3.06	Plasma
Papaioannou, 2011 [58]	UK	Caucasian	33/11	30	28	48	43	30	2	Plasma
Öztürk, 2022 [28]	Turkey	Caucasian	210/62	32.6	30.3	46.4	42.2	31.6	2.8	Serum
Bİçer, 2021 [54]	Turkey	Caucasian	75/75	47	32	29.4	25.9	>5	≤5	Plasma
Ursavas, 2010 [60]	Turkey	Caucasian	55/15	51.1	48.4	32.5	31.6	43.5	2.8	Serum
Gharraf, 2019 [56]	Egypt	Caucasian	30/15	41.63	25.09	51	34.27	43.43	<5	Serum
Garbuio, 2009 [41]	Brazil	Mixed	13/13	37	36	29	27	41	2	Serum

AHI: Apnea–hypopnea index. BMI: Body mass index.

**Table 2 life-13-00149-t002:** Characteristics of the studies reporting before and after CPAP therapy in the meta-analysis.

First Author, Publication Year	Country	Ethnicity	Sample Size	Mean BMI, kg/m^2^	Mean Age, Year	Mean AHI, Events/h		Sample	Follow-Up Duration
						Before CPAP	After CPAP		
Tachikawa, 2016 (i) [39]	Japan	Asian	63	27.9	60.6	42.2	5.6	Plasma	6 weeks
Tachikawa, 2016 (ii) [39]	Japan	Asian	63	27.9	60.6	42.2	3.9	Plasma	3 months
Takahashi, 2008 [42]	Japan	Asian	21	28.5	53.2	39.4	16.1	Plasma	1 months
Yang, 2013 [62]	China	Asian	22	26.7	60	26	3	Plasma	3 months
Sánchez-de-la-Torre, 2012 (iii) [59]	Italy	Caucasian	21	34.34	46.61	48.92	-	Plasma	3 months
Sánchez-de-la-Torre, 2012 (iv) [59]	Italy	Caucasian	28	25.02	34.34	41.45	-	Plasma	3 months
Garbuio, 2009 [41]	Brazil	Mixed	13	37	29	41	4	Serum	6 months
De Santis, 2015 (i) [40]	Italy	Caucasian	11	-	-	-	2.8	Serum	2 days
De Santis, 2015 (ii) [40]	Italy	Caucasian	11	-	-	-	-	Serum	6 months

CPAP: Continuous positive airway pressure. AHI: Apnea–hypopnea index. BMI: Body mass index.

**Table 3 life-13-00149-t003:** The Joanna Briggs Institute (JBI) critical appraisal checklist for case–control studies.

First Author, Publication Year	The Joanna Briggs Institute (JBI) Critical Appraisal Checklist										Quality (Total Quality Score)
	Q1	Q2	Q3	Q4	Q5	Q6	Q7	Q8	Q9	Q10	
De Santis, 2015 [40]	No	Yes	Yes	Yes	Yes	No	No	Yes	Yes	Yes	Moderate (7)
Liu, 2014 [57]	No	Yes	Yes	Yes	Yes	No	No	Yes	Yes	Yes	Moderate (7)
Ciftci, 2005 [55]	No	Yes	No	Yes	Yes	No	No	Yes	Yes	Yes	Moderate (6)
Yang, 2013 [61]	Yes	Yes	Yes	Yes	Yes	No	No	Yes	Yes	Yes	High (8)
Zhang, 2018 [29]	Yes	Yes	Yes	Yes	Yes	No	No	Yes	Yes	Yes	High (8)
Sánchez-de-la-Torre, 2012 (i) [59]	No	No	Yes	Yes	Yes	No	No	Yes	Yes	Yes	Moderate (6)
Sánchez-de-la-Torre, 2012 (ii) [59]	No	No	Yes	Yes	Yes	No	No	Yes	Yes	Yes	Moderate (6)
Papaioannou, 2011 [58]	Yes	Yes	Yes	Yes	Yes	No	No	Yes	Yes	Yes	High (8)
Öztürk, 2022 [28]	No	No	Yes	Yes	Yes	No	No	Yes	Yes	Yes	Moderate (6)
Bİçer, 2021 [54]	Yes	No	Yes	Yes	Yes	No	No	Yes	Yes	Yes	Moderate (7)
Ursavas, 2010 [60]	No	Yes	Yes	Yes	Yes	No	No	Yes	Yes	Yes	Moderate (7)
Gharraf, 2019 [56]	No	No	Yes	Yes	Yes	No	No	Yes	Yes	Yes	Moderate (6)
Garbuio, 2009 [41]	Yes	Yes	Yes	Yes	Yes	No	No	Yes	Yes	Yes	High (8)

Low: 1–4 scores, Moderate: 5–7 scores, High: 8–10 scores.

**Table 4 life-13-00149-t004:** The National Institutes of Health (NIH) quality assessment tool for before–after studies.

First Author, Publication Year	The National Institutes of Health (NIH) Quality Assessment Tool												Quality (Total Quality Score)
	Q1	Q2	Q3	Q4	Q5	Q6	Q7	Q8	Q9	Q10	Q11	Q12	
Tachikawa, 2016 (i) [39]	Yes	Yes	NR	Yes	Yes	Yes	Yes	NR	Yes	Yes	Yes	No	Good (9)
Tachikawa, 2016 (ii) [39]	Yes	Yes	NR	Yes	Yes	Yes	Yes	NR	Yes	Yes	Yes	No	Good (9)
Takahashi, 2008 [42]	Yes	Yes	NR	Yes	No	Yes	Yes	NR	Yes	Yes	Yes	No	Fair (8)
Yang, 2013 [62]	Yes	Yes	NR	Yes	No	Yes	Yes	NR	Yes	Yes	Yes	No	Fair (8)
Sánchez-de-la-Torre, 2012 (iii) [59]	Yes	Yes	Yes	Yes	No	Yes	Yes	NR	Yes	Yes	Yes	No	Good (9)
Sánchez-de-la-Torre, 2012 (iv) [59]	Yes	Yes	Yes	Yes	No	Yes	Yes	NR	Yes	Yes	Yes	No	Good (9)
Garbuio, 2009 [41]	Yes	Yes	Yes	Yes	No	Yes	Yes	NR	Yes	Yes	Yes	No	Good (9)
De Santis, 2015 (i) [40]	Yes	Yes	NR	Yes	No	Yes	No	NR	Yes	No	Yes	No	Fair (6)
De Santis, 2015 (ii) [40]	Yes	Yes	NR	Yes	No	Yes	No	NR	Yes	No	Yes	No	Fair (6)

Good: 9–12 scores, Fair: 5–8 scores, Poor: 1–4 scores. NR = not reported.

**Table 5 life-13-00149-t005:** Subgroup analysis of the correlation between blood levels of ghrelin and several variables in the case–control studies in the meta-analysis.

Variable	Subgroup (N)	SMD	95% CI		*p*-Value	I^2^, %	P_heterogeneity_
			Min.	Max.			
Ethnicity							
	Caucasian (9)	0.32	−0.06	0.70	0.90	80	**<0.00001**
	Asian (3)	0.06	−0.55	0.66	0.85	76	**0.02**
Sample							
	Serum (6)	0.56	0.09	1.03	0.02	79	**0.0003**
	Plasma (7)	0.03	−0.15	0.22	0.73	36	0.15
Sample size							
	≥100 (3)	0.32	−0.67	1.30	0.53	96	**<0.00001**
	<100 (10)	0.28	0.08	0.47	0.005	0	0.58
Mean BMI of adults with OSA, kg/m^2^							
	≥30 (8)	0.19	0.00	0.38	0.05	0	0.54
	<30 (5)	0.36	−0.35	1.08	0.32	91	**<0.00001**
Mean BMI of controls, kg/m^2^							
	≥30 (6)	0.20	−0.02	0.41	0.07	16	0.31
	<30 (7)	0.31	−0.23	0.86	0.26	88	**<0.00001**
Mean age of adults with OSA, year							
	≥45 (7)	0.88	−0.26	0.86	0.30	88	**<0.00001**
	<45 (5)	0.22	0.00	0.44	0.05	22	0.27
Mean age of adults with OSA, year							
	≥45 (3)	0.02	−0.65	0.69	0.95	76	**0.02**
	<45 (9)	0.42	0.04	0.79	0.03	79	**<0.00001**
Mean AHI of adults with OSA, events/h							
	≥40 (7)	0.22	−0.02	0.45	0.07	0	0.50
	<40 (5)	0.41	−0.29	1.11	0.25	91	**<0.00001**
Quality							
	Moderate (9)	0.25	−0.16	0.67	0.23	85	**<0.00001**
	High (4)	0.36	0.04	0.68	0.03	34	0.21

SMD: Standardized mean difference. CI: Confidence interval. BMI: Body mass index. AHI: Apnea–hypopnea index. N: number of studies. Bold number means statistically significant (*p* < 0.05).

**Table 6 life-13-00149-t006:** Subgroup analysis of the correlation between blood levels of ghrelin and several variables in the studies reporting before and after continuous positive airways pressure (CPAP) therapy in the meta-analysis.

Variable	Subgroup (N)	SMD	95% CI		*p*-Value	I^2^, %	P_heterogeneity_
			Min.	Max.			
Ethnicity							
	Caucasian (4)	−0.13	−0.47	0.20	0.43	50	0.11
	Asian (4)	0.15	−0.07	0.36	0.18	0	0.42
Sample							
	Serum (3)	−0.15	−1.25	0.95	0.80	80	**0.007**
	Plasma (6)	0.13	−0.06	0.31	0.19	0	0.66
Sample size							
	≥20 (6)	0.13	−0.06	0.31	0.19	0	0.66
	<20 (3)	−0.15	−1.25	0.95	0.80	80	0.007
Mean BMI, kg/m^2^							
	≥30 (2)	0.33	−0.52	1.17	0.45	65	0.09
	<30 (5)	0.15	−0.05	0.34	0.15	0	0.59
Mean age, year							
	≥45 (5)	0.12	−0.08	0.32	0.24	0	0.52
	<45 (2)	0.34	− 0.10	0.78	0.13	44	0.18
Mean AHI before treatment, events/h							
	≥40 (5)	0.09	−0.11	0.30	0.37	0	0.48
	<40 (2)	0.46	0.03	0.89	0.04	0	0.86
Follow-up duration, month							
	≥3 (6)	0.09	−0.29	0.47	0.63	59	**0.03**
	<3 (3)	0.09	−0.20	0.37	0.55	0	0.46
Quality							
	Good (5)	0.09	−0.11	0.30	0.37	0	0.48
	Fair (4)	−0.02	−0.70	0.65	0.94	71	**0.02**

SMD: Standardized mean difference. CI: Confidence interval. BMI: Body mass index. AHI: Apnea–hypopnea index. N: number of studies. Bold number means statistically significant (*p* < 0.05). OSA: Obstructive sleep apnea.

**Table 7 life-13-00149-t007:** Meta-regression analysis of the correlation between blood levels of ghrelin and several variables in the case–control studies in the meta-analysis.

Variable	Point Estimate	Standard Error	Lower Limit	Upper Limit	Z-Value	*p*-Value
Publication year	0.04139	0.01348	0.01497	0.06780	3.07068	**0.00214**
Sample size	0.00524	0.00100	0.00327	0.00721	5.22125	**<0.00001**
Mean BMI of adults with OSA	−0.00468	0.00894	−0.02221	0.01284	−0.52393	0.60033
Mean BMI of controls	0.01264	0.01316	−0.01316	0.03844	0.96033	0.33689
Mean age of adults with OSA	0.01202	0.00886	−0.00534	0.02938	1.35746	0.17463
Mean age of controls	0.01017	0.00863	−0.00674	0.02708	1.17857	0.23850
Mean AHI of adults with OSA	−0.01818	0.00795	−0.03375	−0.00260	−2.28733	**0.02218**
Quality score	−0.31009	0.09887	−0.50387	−0.11630	−3.13616	**0.00171**

BMI: Body mass index. AHI: Apnea–hypopnea index. Bold number means statistically significant (*p* < 0.05).

**Table 8 life-13-00149-t008:** Meta-regression analysis of the correlation between blood levels of ghrelin and several variables in the studies reporting before and after continuous positive airways pressure (CPAP) therapy in the meta-analysis.

Variable	Point Estimate	Standard Error	Lower Limit	Upper Limit	Z-Value	*p*-Value
Publication year	−0.05749	0.03383	−0.12379	0.00881	−1.69959	0.08921
Sample size	−0.00738	0.00456	−0.01631	0.00155	−1.61907	0.10543
Mean BMI	0.01806	0.03137	−0.04342	0.07954	0.57583	0.56473
Mean age	−0.00853	0.00883	−0.02583	0.00877	−0.96615	0.33397
Mean AHI before treatment	−0.02764	0.01775	−0.06242	0.00714	−1.55739	0.00938
Quality score	0.15786	0.10484	−0.04763	0.36334	1.50569	0.13215

AHI: Apnea–hypopnea index. BMI: Body mass index.

## Data Availability

No new data were created or analyzed in this study. Data sharing is not applicable to this article.

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
