# Peer review of "Effect of Continuous Positive Airway Pressure on Changes of Plasma/Serum Ghrelin and Evaluation of These Changes between Adults with Obstructive Sleep Apnea and Controls: A Meta-Analysis"

_life, 2023, doi:10.3390/life13010149_

Round 1

Reviewer 1 Report

The authors have conducted a meta-analysis that consists of two parts. They have investigated whether adults with obstructive sleep apnea (OSA) have a higher level of ghrelin in plasma/serum and whether continuous positive airway pressure therapy (CPAP) would change the plasma/serum level of ghrelin in adults with obstructive sleep apnea. To confirm the controversial association of OSA and ghrelin level, this meta-analysis has summarized the currently available evidence and provided a conclusion pointing out that the association between OSA and blood ghrelin level, and the effects of CPAP on changing the blood ghrelin level are not significant. However, the manuscript can be improved by addressing the following comments.

There are two forms of ghrelin, acylated form, and unacylated form. While mentioning the blood level of ghrelin, is it referred to the total ghrelin, acylated ghrelin, or unacylated ghrelin? Does this meta-analysis only include studies with measurement of total ghrelin, or it includes studies that investigate the change in any form of ghrelin? Further clarification is needed to avoid confusion.

The authors need to elaborate more on why ghrelin is an important target in the introduction. Were the authors trying to link up other health conditions with OSA and believe the level of ghrelin is the mediator? What health condition is related to the upregulation of ghrelin? Is that the patients would be more susceptible to some diseases or disorders if the ghrelin level was increased due to the presentation of OSA?

The paragraph on Page 2 line 78 is mentioning the rationale of conducting this study. However, the paragraph did not highlight the advancement of the current study compared with the previous meta-analysis published in 2021. While only two studies were published on or after 2021, why the current meta-analysis contains more studies published before 2021 than the previous one? Is that because of the choices of database or the searching criteria? The authors also mentioned that the current study contains additional analysis compared with the previous one. Why are these additional analyses important? What values can these additional analyses add to this study and make it more advance than the previous one?

How many independent reviewers were involved in this study?

In Page 3 line 100: “… were checked to no study was missed”, please amend to “… were checked to  ensure no study was missed”,

The format of Table 1 and 2 require some tidy-up.

Pare 8 line 207, the purposes for performing such subgroup analyses should be clarified. Why were those variables selected as the parameters for the subgroup analyses?

Table 6 and 7 need to be tidied up. “Ethnicity” was in the labelling row in both tables.

Page 14 line 293, please mention the results after removing outliers was a sensitivity analysis.

The discussion section did not discuss the additional analyses which are supposed to be the advancement of the current study compared with the previous one. Please discuss the results of those additional analyses and what values were added.

What does the significant results of sample type and sample size in table 6 imply?

What does the significant result of the publication year in the meta-regression in table 8 imply?

What does the significant result of the quality in the meta-regression in table 8 imply?

I did not see the purpose of the paragraph on page 15 line 322. Why did the authors suddenly discuss HDL? The authors mentioned that mixed results on the association between HDL and OSA were obtained. Can the authors clarify what values this paragraph adds to the discussion and why it is important? I think it is outside the scope of this study.

Please mention the results of this current study are in line with the previously published meta-analysis

Author Response

We thank Reviewer #1 for their valuable comments, which helped us to improve the quality of the present revision. Please find attached the detailed point-by-point-response. Again, thank you very much for the care devoted to our work.

Reviewer 2 Report

Thank you for giving me to check your meta-analysis review titled 'Effect of continuous positive airway pressure on changes of plasma/serum ghrelin and evaluation of these changes between adults with obstructive sleep apnea and controls'. It's very interesting for me about gherelin. I'm not familiar to ghrelin. As you know, ghrelin is a hormone produced by enteroendocrine cells of the gastrointestinal tract, especially the stomach, and is often called a "hunger hormone" because it increases the drive to eat. I think it is related to obesity and OSA with CPAP therapy, and your focus was very good point. Unfortunately, in this meta-analysis,  there were no significant difference between gherelin and OSA, but, currently, gherelin itself was not well measured, but further detailed studies focusing on ghrelin would clarify the relationship between OSA and ghrelin. 

Author Response

We thank Reviewer #2 for their valuable comments, which helped us to improve the quality of the present revision. Please find attached the detailed point-by-point-response. Again, thank you very much for the care devoted to our work.

Reviewer 3 Report

I have read the systematic review entitled ‘Effect of continuous positive airway pressure on changes of plasma/serum ghrelin and evaluation of these changes between adults with obstructive sleep apnea and controls: A meta-analysis’ with great interest (manuscript ID life-2067014).

Authors of publication aimed to investigate changes in serum/plasma levels of ghrelin in adults with OSA compared to controls and before/after continuous positive airway pressure (CPAP) therapy in adults with OSA. 

They included n=362potentially relevant articles into synthesis following the PRISMA Statement guidelines.

Authors searched articles from PubMed, Web of Science, Scopus, and Cochrane Library without any restrictions, and after thorough analysis they described 13 studies for case-control studies and 9 articles for before-after studies.

The paper has some strengths: cautious in not overinterpreting findings, conclusive summary and future attempts.

However, there were some editorial and substantive gaps, I would like to address:

How many authors performed the literature review? Please report.

Please report when did you start your database search.

lines 161-162 At last, 15 articles involving 13 studies for case-control studies and 9 articles for before-after studies were included. - please rewrite the sentence, it seems to be miscounted

I have not assessed statistics.

After minor corrections, I suggest Editor this paper for publication.

Author Response

We thank Reviewer #3 for their valuable comments, which helped us to improve the quality of the present revision. Please find attached the detailed point-by-point-response. Again, thank you very much for the care devoted to our work.

Round 2

Reviewer 1 Report

The authors have addressed my comments. With some English language editing, the manuscript can be accepted.